# Endocrine-Disrupting Chemicals and the Development of Diabetes Mellitus Type 1: A 5-Year Systematic Review

**DOI:** 10.3390/ijms251810111

**Published:** 2024-09-20

**Authors:** Georgia-Nektaria Keskesiadou, Sophia Tsokkou, Ioannis Konstantinidis, Maria-Nefeli Georgaki, Antonia Sioga, Theodora Papamitsou, Sofia Karachrysafi

**Affiliations:** 1Research Team “Histologistas”, Interinstitutional Postgraduate Program “Health and Environmental Factors”, Department of Medicine, Faculty of Health Sciences, Aristotle University of Thessaloniki, 54124 Thessaloniki, Greece; gwgwkeske@gmail.com (G.-N.K.); stsokkou@auth.gr (S.T.); ikonsc@auth.gr (I.K.); mgeorgaki@cheng.auth.gr (M.-N.G.); asioga@auth.gr (A.S.); sofia_karachrysafi@outlook.com (S.K.); 2Laboratory of Histology-Embryology, Department of Medicine, Faculty of Health Sciences, Aristotle University of Thessaloniki, 54124 Thessaloniki, Greece; 3Environmental Engineering Laboratory, Department of Chemical Engineering, Aristotle University of Thessaloniki, 54124 Thessaloniki, Greece

**Keywords:** Endocrine Disrupting Chemicals (EDCs), Diabetes Mellitus Type 1 (T1DM), phthalates, Persistent Organic Pollutants (POPs), bisphenol A (BPA)

## Abstract

Introduction: According to the Institute of Environmental Sciences, endocrine-disrupting chemicals (EDCs) are “natural or human-made chemicals that may mimic, block, or interfere with the body’s hormones, associated with a wide array of health issues”, mainly in the endocrine system. Recent studies have discussed the potential contribution of EDCs as risk factors leading to diabetes mellitus type 1 (T1DM), through various cellular and molecular pathways. Purpose: The purpose of this study was to investigate the correlation between the EDCs and the development of T1DM. Methodology: Thus, a 5-year systematic review was conducted to bring light to this research question. Using the meta-analysis and systematic review guideline protocol, a PRISMA flow diagram was constructed and, using the keywords (diabetes mellitus type 1) AND (endocrine-disrupting chemicals) in the databases PubMed, Scopus and ScienceDirect, the relevant data was collected and extracted into tables. Quality assessment tools were employed to evaluate the quality of the content of each article retrieved. Results: Based on the data collected and extracted from both human and animal studies, an association was found between T1DM and certain EDCs, such as bisphenol A (BPA), bisphenol S (BPS), persistent organic pollutants (POPs), phthalates and dioxins. Moreover, based on the quality assessments performed, using the Newcastle–Ottawa Scale and ARRIVE quality assessment tool, the articles were considered of high quality and thus eligible to justify the correlation of the EDCs and the development of T1DM. Conclusion: Based on the above study, the correlation can be justified; however, additional studies can be made focusing mainly on humans to understand further the pathophysiologic mechanism involved in this association.

## 1. Introduction

### 1.1. Endocrine Disrupting Chemicals

Endocrine-disrupting chemicals (EDCs) are substances present in the environment such as soil, air and water and in food sources. They can also be found in manufactured and cosmetic care products [1]. They are found in their natural form or are human-produced and have the tendency to mimic, inhibit and intervene in the natural pathway of the human body’s hormones and thus lead to a wide range of health pathological alterations [2]. Generally, EDCs can act in three main ways. They can act by mimicking human hormones and inhibiting the natural purpose of a hormone in the physiologic pathway. The second way of interfering is either by increasing or decreasing the hormones by affecting their metabolic and storing processes. Thirdly, ECDs can affect the level of sensitivity the human body has to the various types of hormones. These kinds of alterations can lead to several adverse effects on the health of an individual, such as diabetes mellitus (DM), obesity and cardiovascular diseases (CVD), as well as growth restrictions and neurological and cognitive disabilities [1,2]. EDCs are thoroughly distributed in our daily life and are found in daily products such as cosmetics, seafoods, pesticides and packaging of children’s toys [2].

### 1.2. Diabetes Mellitus

Diabetes mellitus (DM) is a group of metabolic diseases characterized by chronic hyperglycemia due to disturbances in insulin secretion, insulin action in the form of resistance, or a combination of these two. Inadequate insulin secretion and/or reduced tissue response to insulin results in insufficient action of insulin on the targeted tissues, leading to disruptions of carbohydrate, fat and protein metabolism. Both decreased insulin secretion and insulin action may coexist in the same patient. While the etiology of diabetes is heterogeneous, most cases of DM can be classified into two broad etiopathogenetic categories. The first type is T1DM, which is mainly characterized by insulin secretion deficiency, while T2DM results from a combination of resistance to insulin action and the insufficient compensatory insulin secretion response. While T1DM remains the most common form of diabetes in young people in many populations, especially those with a European background, T2DM is becoming an increasingly important public health concern worldwide, among children as well as in people with severe obesity [1,2].

#### 1.2.1. Diabetes Mellitus Type 1

More specifically, T1DM, also known as juvenile diabetes or insulin-dependent diabetes, has been of particular concern in recent years, as the number of cases is ever increasing, and is the most common type in children and adolescents. The Atlas of Diabetes (7th edition) reports that its global prevalence is estimated at 415 million (8.8%), which is projected to increase to 642 million over the next 25 years. In India, there are about 69.2 million people with diabetes, but this figure is expected to reach 123.5 million by 2040 [3]. The reason for the increase in the number of cases is still unclear, but there are several risk factors that lead to predisposition to the development of T1DM, including environmental factors, genetic factors, epigenetic factors, lifestyle and complications during pregnancy, such as preeclampsia and infections [4,5].

#### 1.2.2. Diabetes Mellitus Type 1 and EDCs (Figure 1)

Research that is based on the environmental factors playing a role in the etiology of pancreatic islet autoimmunity focus mainly on the impact of viruses, inappropriate early infant nutritional diet, high childhood weight gain, vitamin D deficiency, gut microbiome and endocrine-disrupting chemicals (EDCs) [6].

EDCs are a group of exogenous compounds with high heterogeneity that can be found naturally in living organisms or are synthesized industrially in food and consumer products [6,7]. They can influence the development and function of beta cells or immune system genes, promoting autoimmunity and increasing susceptibility to autoimmune attack as -individual chemical agents or as chemical mixtures. However, there are only a few conflicting human studies showing the possible role of exposure to EDCs in the pathogenesis of T1DM. In addition, the presence of a familial form of mild diabetes during adolescence should raise the suspicion of monogenic diabetes, which is responsible for 1% to 6% of cases of childhood DM.

Furthermore, there is increasing evidence indicating a connection between exposure to endocrine-disrupting chemicals (EDCs) and the development of various pancreatic diseases, including type 1 diabetes mellitus (T1DM). This correlation is linked to the aryl hydrocarbon receptor (AHR), which acts as a transcription factor that is activated by ligands. The AHR plays a critical role in regulating essential cellular and molecular processes, such as the metabolism of foreign substances, immune responses and the formation of cancer. Notably, in the developing pancreas of the embryo and in the early stages of life, a variety of endocrine-disrupting substances (EDCs) function as agonists or antagonists of AHR, causing dysregulation of critical cellular and molecular pathways and disturbance of the endocrine system [8]. 

**Figure 1 ijms-25-10111-f001:**
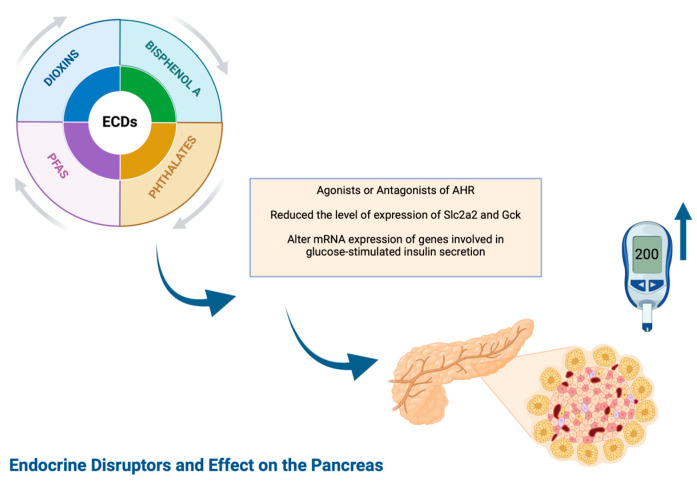
Endocrine disruptors and effect on the pancreas, created by BioRender.

#### 1.2.3. Epidemiology

According to the International Diabetes Federation (IDF), 1 in 10 adults have DM worldwide, while 3 in 4 people with DM live in low-income countries and 1 in 2 patients with DM are not diagnosed. T1DM affects 1.2 million children worldwide. Moreover, according to the IDF, there is an exponential increase in cases, especially in regions of the world that are of lower socioeconomic status [9].

#### 1.2.4. Stages of Diabetes Mellitus Type 1

T1DM is characterized by three stages. More precisely, at stage 1, the presymptomatic stage, there are autoantibodies against islets (two or more types), and normal blood glucose. The immune system at this stage has already begun to attack the beta cells of the pancreas; however, blood sugar remains normal and symptoms do not yet appear. Subsequently, in stage 2, in addition to the presence of autoantibodies there is abnormal glucose tolerance, usually being presymptomatic, due to increasing beta cell loss/destruction. For both stages (stage 1 and stage 2) the risk of developing T1DM approaches its peak. Finally, in stage 3, the clinical diagnosis of T1DM takes place, as blood glucose is now above diagnostic limits, while at the same time the common symptoms of polyuria, polydipsia, weight loss and fatigue appear. Clinical research supports the necessity of early diagnosis of T1DM before stage 3, as the sooner the diagnosis is made, the sooner the intervention will be made, reducing the occurrence of long-term complications [5,6,7].

#### 1.2.5. Diagnostic Criteria of T1DM (Figure 2)

According to the World Health Organization (WHO), the presence of even a random value of fasting plasma glucose ≥200 mg/dL in combination with the characteristic clinical symptoms of diabetes make the diagnosis definitive. In addition, a fasting glucose value of ≥126 mg/dL, as long as fasting lasts at least 8 h before the test, as well as the absence of symptoms confirmation with a second test or a second sample, can result in a diagnosis of diabetes. Another method employed to diagnose diabetes in recent years is glycosylated hemoglobin A1c (HbA1c), a widely used marker for chronic glycemia that measures non-enzymatic glycation of hemoglobin and reflects average blood glucose levels over a period of 2 to 3 months. If it is at an HbA1c level of ≥6.5% (according to an estimated average glucose of 140 mg/dL), then the diagnosis is made again. Finally, there is the oral glucose tolerance test (OGTT), which most of the time—especially for the diagnosis of T1D in children and adolescents—is not necessary, but in cases of asymptomatic hyperglycemia and investigation of abnormal glucose metabolism outside T1D the test is useful and should be applied when appropriate. OGTT is performed after loading with 1.75 g of glucose per kg of body weight (BS), with a maximum dose of 75 g of glucose. The diagnosis, therefore, is definitive when the blood glucose value two hours after this test is ≥200 mg/dL [1,2]. 

**Figure 2 ijms-25-10111-f002:**
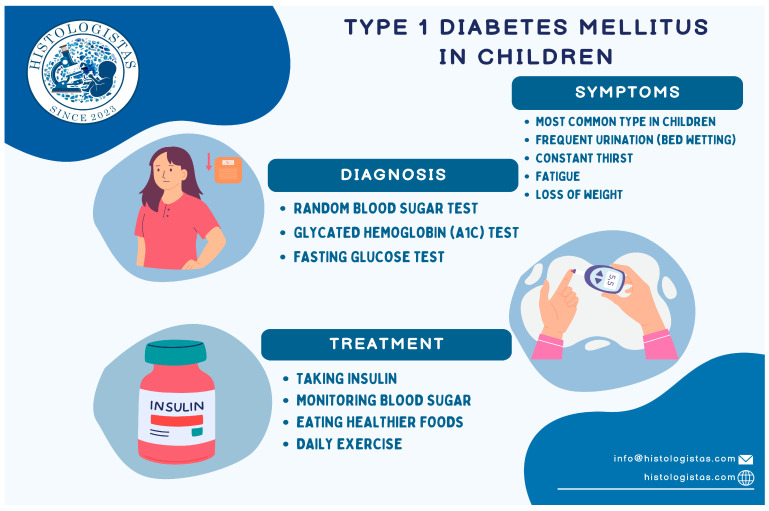
Symptoms, diagnostic criteria and treatment of T1DM.

#### 1.2.6. Basic Treatment Options for T1DM (Figure 2)

Regarding basic treatment options for T1DM, the initiation of subcutaneous insulin is of utmost importance, in combination with intensive blood glucose monitoring, adoption of a healthy diet and daily exercise. Many regimens of insulin injection intensity have been proposed, with the main notion being the utilization of both a long-acting insulin (basal insulin) and a rapid-acting insulin before meals [1,2].

## 2. Methods 

### 2.1. Purpose of Study

The purpose of this study was to investigate the correlation between the development of T1DM and EDCs. Thus, a 5-year systematic review was constructed to answer the question raised.

### 2.2. PICO Table–PRISMA Flow Diagram–Table–Quality Assessment

The Population, Intervention, Control and Outcome (PICO) table was created to facilitate and clarify the question more clearly. In addition, a PRISMA flow diagram was produced using the following keywords in the code: (diabetes mellitus type 1) AND (endocrine disruptors). Then the databases PubMed, Scopus and Science Direct were searched with the following criteria: duration in the last 5 years (2019–2024), the articles had to be Reviews, Research, Articles and Case Reports, written in English and have open access. Tables were then constructed with the most crucial information retrieved (Cohort Studies, Basic Research, Literature Review) and quality assessments were performed (Newcastle–Ottawa Scale, ARRIVE quality assessment tool) to check the accuracy of the papers. The Newcastle–Ottawa Scale assessment is for cohort studies, with a scale of 0 (lowest) to 7 (highest). The second rating scale used was the ARRIVE scale, which is specific to animal studies and ranges from 1 to 10. 

## 3. Results

### 3.1. PICO Table

According to the PICO, the study population consisted of children, adolescents, and young adults (10 to 22 years old), the risk factor was endocrine disruptors, as an incidence of T1DM and as a result endocrine disruptors as risk factors for developing T1DM (Table 1). 

### 3.2. PRISMA Flow Diagram

After the PRISMA flow diagram was created, the following results were found. A total of 17,947 articles were found using the keywords (diabetes mellitus type 1) AND (endocrine disruptors). Specifically, 37 articles were found in PubMed, 205 in Scopus and 17,705 in Science Direct. After using the criteria, which are: in the last 5 years (2019–2024), the articles had to be Reviews, Studies, Articles and Case Reports, written in English and have open access, the automatic data engine removed 16,413 articles, leaving 1534 articles remaining. These articles were studied, and 14 articles were removed as duplicates, leaving 1520. From these, a total of 1505 articles were removed, specifically 189 articles were removed due to low correlation based on title, 634 based on abstract, 671 based on total text and, finally, 11 since only abstracts were found. A total of 16 articles were selected from the databases. An additional 6 articles from external sources were added to the total number, bringing the final number to 22 articles (Figure 3). Three tables were then created that included the most important information, more specifically, the names of the authors, the chronology and type of articles, the groups with the highest correlation with T1DM, the types of endocrine disruptors and their correlation with T1DM (Table 2, Table 3 and Table 4).

### 3.3. Quality Assessment

#### 3.3.1. Newcastle–Ottawa Scale (NOS)

Firstly, the Newcastle–Ottawa Scale (NOS) was used to evaluate four nonrandomized trials, of which two of the four, namely Salo HM, 2019 [10] and Lee, I, 2021 [11] had a score of 7, which is the highest score, followed by the Bresson SE 2019 [12] study, with a score of 6, and, finally, the Duforun study, P, 2023 [13] with a score of 5. The last two studies were on the middle of the scale, but the overall score of all four papers can support the credibility of the overall content (Table 5).

#### 3.3.2. ARRIVE Quality Assessment Scale 

In addition, the ARRIVE scale used for animal studies showed the following results. Xu’s study, J, 2019 [14] had the highest score of 9, while the remaining 4 studies had scores of 7, which is on the medium to high scale (Table 6).

**Table 2 ijms-25-10111-t002:** Cohort studies.

Authors	Study Type	Type of ECDS	ECDS Correlation with T1DM	Number of Population	Method of Assessment	Risk Factors	Sex	Age
Salo HM, 2019 [10]	Cohort	1. Persistent organic pollutants (POPs) 2. dioxins 3. polychlorinated biphenyls (PCBs) 4. Pesticides 5. Brominated flame-retardants	Could not observe any definite associations between increased exposure to chemical pollutants at birth, at 12 or at 48 months of age, and risk of β-cell autoimmunity. The current work indicates that prenatal or early childhood exposure to POPs, including PFASs, is not an apparent risk factor for later β-cell autoimmunity. In 48-month-old children, PFDA was above the LOQ in 34% of the autoantibody-negative children, in 63% of the autoantibody-positive children, and in 88% of the children diagnosed with type 1 diabetes. PFDA has been demonstrated to interfere with the function of thyroid hormones in in vitro studies, and endocrine disruption is an interesting mode of action of PFDA in biological systems.	136	1. FINDIA pilot study 2. DIABIMMUNE study	Autoantibody positive cases, HLA risk genotype, Breastfeeding, Formula feeding	XX, XY	from newborns to 6 years old
Bresson SE, 2019, [12]	Cohort	1. Polychlorinated biphenyls (PCBs) 2. Organochlorine pesticides 3. POPs	There was a significant association between T1D/IS and several POPs including p, p’-DDE, trans-nonachlor and PCB-153. PCB-153 and p, p’-DDE reduce insulin production and secretion in β-cells. Interestingly, PCB-153 but not p, p’-DDE, reduced the expression level of Slc2a2 and Gck. PCB-153 and p, p’-DDE alter mRNA expression of genes involved in glucose-stimulated insulin secretion.	442	SEARCH Case-Control (SEARCH-CC)	BMI, Insulin Sensitivity, Insulin Resistance	XX, XY	10 to 22 years
Dufour, P, 2023 [13]	Cohort	7 phthalate metabolites, 4 parabens, 7 bisphenols, benzophenone 3 and triclosan were measured in urine, while 15 organochlorine pesticides, 4 polychlorinated biphenyls (PCBs) and 7 perfluoroalkyl substances 1. Phthalate metabolites 2. PCBs 3. Bisphenols 4. PCB 5. Miscellaneous, OCP, PFAS	This work investigated the link between the exposure to some environmental pollutants, from persistent organic pollutants to some non-persistent plasticizers and antimicrobials, and thyroid disorders in type 1 diabetes children. Associations between the levels (in serum or urine) of some PFASs, PCBs, phthalates and bisphenols and thyroid hormone were highlighted, suggesting an impact of these pollutants on the thyroid function in this population suspected to be particularly vulnerable toward endocrine disruption.	54	Urine and Blood Analysis	-	XX, XY	3 to 18 years
Lee, I, 2021 [11]	Cohort	Phthalates and bisphenol A (BPA), parabens	BPA in urine was associated with higher rates of DM	3787	1. Korean National Environmental Health Survey (KoNEHS) 2. Concentrations of phthalate metabolites, BPA, and parabens were measured in the spot urine samples using liquid–liquid extraction and ultraperformance liquid chromatography mass spectrometry separation, followed by electrospray ionization and tandem mass spectrometry 3. Urinary dilution, in addition to two traditional methods (i.e., Cr adjustment and SG adjustment), a CAS was used	-	XX, XY	19 and older

**Table 3 ijms-25-10111-t003:** Basic Research–Animal Studies.

Authors	Type of Study	Type of ECDS	ECDS Correlation with T1DM	Number of Population	Method of Assessment	Sex	Age
Xu J, 2019 [14]	Basic Research Study	Bisphenol A (BPA)	Not statistically significant, but a shift towards accelerated T1D development was observed	30	BW and non-fasting BGLs were measured every 1–2 weeks. Accu-Chek Diabetes monitoring kit (Roche Diagnostics, Indianapolis, IN, USA) or Contour Blood Glucose Meter (Ascensia Diabetes Care, Parsippany, NJ, USA) were used to measure BGLs from a small sample of venous blood (tail nick).	XX	8 to 12 weeks old
McDonough CM, 2022 [15]	Basic Research Study	Bisphenol S (BPS)	Significant adverse effect	12	Body weight, blood glucose measurement, diabetic incidence, GTT and ITT, behaviour test, Y-maze test, Flow Cytometric Analysis	XX, XY	10 weeks
Sinioja T, 2022 [16]	Basic Research Study	Persistent organic pollutants (POPs), organochlorides, organobromides, and per- and polyfluoroalkyl substances (PFAS)	Risk of T1D	29	Folch procedure for Lipidomic Analysis	XX, XY	8 to 10 weeks old
Xu J, 2019 [17]	Basic Research Study	Bisphenol A (BPA)	Sex plays an important role in BPA altering T1D risk. BPA accelerated T1D development in adult NOD females but delayed T1D development in male mice.	-	Tolerance tests and insulin measurement, antibody measurement, cytokine/chemokine measurement, GMB, bioinformatics and metabolomics and bioinformatics analysis	XX, XY	8 to 12 weeks old
Xu J, 2019 [18]	Basic Research Study	Bisphenol S (BPS)	PS exposure had dose-related protective effects on T1D in females. This suggests that BPS uses different mechanisms from BPA to alter glucose homeostasis and T1D	-	Tolerance tests and insulin measurement, antibody measurement, cytokine/chemokine measurement, GMB, bioinformatics and metabolomics and bioinformatics analysis		8 to 15 weeks old

**Table 4 ijms-25-10111-t004:** Literature reviews.

Authors	Study Type	Type of ECDS	ECDS Correlation with T1DM	Doi
Melissa and Cairro, 2023 [19]	Review	Phthalates	There is a possible link between the exposure to phthalates and the development of DM. Phthalates may lead to insulin resistance and consequent diabetes mellitus through oxidative stress, the activation of different hormone receptors (PPAR and ER), and impaired inflammatory factors.	10.3390/metabo13060746
Prediere et al., 2020 [20]	Review	Chemical substances (bisphenol A; pesticides; phthalates; polychlorinated biphenyls; polyfluorinated substances)	They could affect the development and the function of the immune system or of the β-cells function, promoting autoimmunity and increasing the susceptibility to autoimmune attack. However, the studies are few and demonstrated contradictory results, according to gender and age.	10.3390/ijms21082937
HInault et al., 2023 [21]	Review	Organochlorine (OCs) pesticides: dichloro-diphenyl-trichloroethane (DDT) and its metabolites (chlordane) and industrial chemicals: polychlorinated biphenyls (PCBs), polybrominated diphenyl ethers (PBDE), dioxins (TCDD: 2,3,7,8-tétrachlorodibenzo-p-dioxine), and per- and polyfluoroalkyl substances (PFAs), bisphenol A (BPA)	It remains difficult to form a comprehensive view on the causal relationship between EDCs and diabetes (both T1DM and T2DM), and further experiments are required.	10.3390/ijms24054537
Khali et al., 2023 [22]	Review	Ambient air pollution, persistent organic pollutants (POPs), metals (bisphenol A [BPA]), phthalates, polybrominated diphenyl ethers (PBDEs)	The research has shown inconsistent results regarding direct pathogenesis, since these diseases are multifactorial. Many studies attempted to determine the impact of isolated atmospheric compounds but did not taking into consideration their compounding effects, and others used smaller sample sizes.	10.3390/ijms24108870
Heo and Kim, 2021 [23]	Review	Ambient air pollution (PM, NO_2_, and NOx)	Altered immune response, oxidative stress, neuroinflammation, inadequate placental development, and epigenetic modulation are some of the underlying factors that have been identified. However, it is difficult to demonstrate causality.	10.6065/apem.2142132.066
Jiang et al., 2023 [24]	Review	bisphenol A (BPA)	BPA exposure is associated with target organ damage in DM and may exacerbate the progression of some chronic complications of DM.	10.1016/j.heliyon. 2023.e16340
Ibarra et al., 2020 [25]	Review	bisphenol A (BPA) & phthalates	There is association with a wide range of reproductive, metabolic and neurological diseases, as well as hormone-related cancers.	10.1016/j.envpol.2020.116380

**Table 5 ijms-25-10111-t005:** Newcastle–Ottawa Quality Assesment Scale–Cohort Studies.

	Selection				Comparability	Outcome		
Source	Representativeness of the Sample	Sample Size	Non–Respondents	Ascertainment of the Exposure	The Subjects in Different Outcome Groups Are Comparable	Assessment of the Outcome	Statistical Test	Quality
Salo HM, 2019 [10]	1	1	1	1	1	1	1	7
Bresson SE, 2019, [11]	1	1	1	1	0	1	1	6
Lee, I, 2021 [12]	1	1	1	1	1	1	1	7
Dufour, P, 2023 [13]	1	1	1	1	0	0	1	5

Methodological quality according to total score: <5: low quality, 5–7: moderate quality, >7: high quality.

**Table 6 ijms-25-10111-t006:** ARRIVE assessment scale—animal studies.

Source	Study Design	Sample Size	Inclusion and Exclusion Criteria	Randomisation	Blinding	Outcome Measures	Statistical Methods	Experimental Animals	Experimental Procedures	Results	Quality
Xu J, 2019 [14]	+	+	+	+	?	+	+	+	+	+	9
McDonough CM, 2022 [15]	+	+	?	?	?	+	+	+	+	+	7
Sinioja T, 2022 [16]	+	+	-	+	?	?	+	+	+	+	7
Xu J, 2019 [17]	+	-	-	+	?	+	+	+	+	+	7
Xu J, 2019 [18]	+	-	-	+	?	+	+	+	+	+	7

“?” = uncertain, not stated; “+” = positive answer to assessment questions; “-“= negative answer to assessment questions.

## 4. Discussion

Based on the systematic research conducted, data collected using the PRISMA guidelines and PICO model, the following results can be stated from the information summarized in the three tables that, respectively, represent cohorts carried out with human participants (Table 2), basic research studies conducted with animals (Table 3) and, finally, literature and systematic reviews (Table 4).

### 4.1. Correlation between ECDs and T1DM Based on Human Cohort Studies

Firstly, two major retrospective cohort studies discussed the association of polychlorinated biphenyls (PCBs), organochlorine pesticides and persistent organic pollutants (POPs) with the descriptive results of the studies mentioned in Table 2. In Bresson SE et al., 2019, where 442 people, both males and females, of age 10 to 20 years old were tested according to the SEARCH Case-Control protocol [11], there was a significant association between T1DM/IS occurrence and several POPs, including p, p’-DDE, trans-nonachlor and PCB-153. PCB-153 and p, p’-DDE were found to reduce insulin production and secretion in pancreatic β-cells. Interestingly, PCB-153 but not p, p’-DDE reduced the level of expression of *Slc2a2* and *Gck*, but both were confirmed to alter mRNA expression of genes involved in glucose-stimulated insulin secretion. Additionally, they identified increased BMI, reduced insulin sensitivity and insulin resistance as potential risk factors for ECDs to cause T1DM. 

However, Salo HM et al., 2019 [10], where 136 children, both male and female, from newborn to 6 years old were studied according to FINDIA pilot study and DIABIMMUNE study protocols [1], could not observe any definite associations between increased exposure to chemical pollutants at birth, or at 12 or at 48 months of age, and risk of β-cell autoimmunity and, thus, the development of T1DM. Prenatal or early childhood exposure to POPs, including PFASs, is not an apparent risk factor for later β-cell autoimmunity, but in 48-month-old children, PFDA was above the LOQ in 34% of the autoantibody-negative children, in 63% of the autoantibody-positive children, and in 88% of the children diagnosed with type 1 diabetes. PFDA has been demonstrated to interfere with the function of thyroid hormones in in vitro studies, and endocrine disruption is an interesting mode of action of PFDA in biological systems. Autoantibody positive cases, HLA risk genotype, breastfeeding and formula feeding were observed as latent risk factors.

Secondly, phthalates, bisphenol A (BPA) and parabens were investigated as impending ECDs in the pathophysiology of T1DM in Lee I. et al., 2021 [12]. They studied 3787 participants older than 19 years old, both male and female, utilizing the Korean National Environmental Health Survey (KoNEHS), concentrations of phthalate metabolites, BPA, and parabens that were measured in spot urine samples using liquid–liquid extraction and ultraperformance liquid chromatography mass spectrometry separation, followed by electrospray ionization and tandem mass spectrometry and urinary dilution, in addition to two traditional methods (i.e., Cr and SG adjustment). This cohort concluded that increased BPA in urine was associated with higher rates of T1DM.

Moreover, Dufour P. et al., 2023 [13] analyzed blood and urine samples of 54 children aged 3 to 18 years old, both male and female, in an attempt to investigate the link between the exposure to many environmental pollutants, from persistent organic pollutants to some non-persistent plasticizers and antimicrobials [i.e., 7 phthalate metabolites, 4 parabens, 7 bisphenols, benzophenone 3, triclosan, 15 organochlorine pesticides, 4 polychlorinated biphenyls (PCBs) and 7 perfluoroalkyl substances] and thyroid disorders in T1DM children. Associations between the levels (in serum or urine) of some PFASs, PCBs, phthalates and bisphenols and thyroid hormone levels were highlighted, suggesting an impact of these pollutants on thyroid function in this population, suspected to be particularly vulnerable toward endocrine disruption, but there was no direct implication that they correlate with the development of T1DM.

### 4.2. Correlation between ECDs and T1DM Based on Basic Research Animal Studies

Furthermore, four different study protocols by Xu J. et al., 2019 [14,16,17,18], using non-obese diabetic (NOD) mice, male and/or female, aged 8 to 12 weeks old, rummaged the association of bisphenol A (BPA) and bisphenol S (BPS) in T1DM occurrence. In the first experiment, BW and non-fasting BGLs were measured every 1–2 weeks using an Accu-Chek Diabetes monitoring kit (Roche Diagnostics, Indianapolis, IN, USA) or Contour Blood Glucose Meter (Ascensia Diabetes Care, Parsippany, NJ, USA) from a small sample of venous blood (tail nick), and it revealed a not statistically significant correlation, but a shift towards accelerated T1DM development was observed. The other two studies assessed NOD mice with tolerance tests and insulin, antibody, cytokine/chemokine measurement, GMB, bioinformatics and metabolomics analysis and verified that sex plays an important role in BPA altering T1DM risk, as BPA accelerated T1DM development in adult NOD females, but delayed male mice from T1DM development, and, also, that BPS exposure had dose-related protective effects on T1DM in females, suggesting that BPS uses different mechanisms from BPA to alter glucose homeostasis and T1DM. In addition, McDonough CM et al., 2022 [15] assessed 10-week-old NOD mice using body weight, blood glucose measurement, diabetic incidence, GTT and ITT, behavior tests, the Y-maze test and Flow Cytometric Analysis to substantiate that BPS has a significant adverse effect on T1DM occurrence. The last animal study to examine the correlation between ECDs and T1DM development risk was by Sinioja T et al., 2022 [16], where they attested that persistent organic pollutants (POPs), organochlorides, organobromides and per- and poly-fluoroalkyl substances (PFAS) are risk factors for the development of T1DM, using 29 8- to 10-week-old NOD mice and assessing them with the Folch procedure for Lipidomic Analysis.

### 4.3. Correlation between ECDs and T1DM Based on the Literature and Systematic Reviews

Many reviews tried to support the association between different disruptors and the development of T1DM. Firstly, Melissa and Cairro, 2023 [19] stated that there is a possible link between the exposure to phthalates and the development of DM, as they may lead to insulin resistance and consequent DM through oxidative stress, the activation of different hormone receptors (PPAR and ER) and impaired inflammatory factors. Secondly, Ibarra et al., 2020 [25] asserted that there is an association of bisphenol A (BPA) and phthalates with a wide range of reproductive, metabolic and neurological diseases, as well as hormone-related cancers, while Jiang et al., 2023 [24] declared that BPA exposure is associated with target organ damage in DM and may exacerbate the progression of some chronic complications of DM. Moreover, Prediere et al., 2020 [20] averred that many chemical substances (Table 4) could affect the development and the function of the immune system or of the pancreatic β-cells, promoting autoimmunity and increasing the susceptibility to autoimmune assault, but the studies cited are few and demonstrated contradictory results, according to gender and age. Furthermore, Heo and Kim, 2021 [23] testified that altered immune response, oxidative stress, neuroinflammation, inadequate placental development and epigenetic modulation are some of the underlying risk factors that have been identified for the pathophysiology of ambient air pollution (i.e., PM, NO_2_ and NOx) in the development of T1DM. Additionally, Khali et al., 2023 [22] expressed that current research has shown inconsistent results with regard to direct pathogenesis, due to the fact that DM is multifactorial and many studies attempted to determine the impact of isolated atmospheric compounds, such as ambient air pollution, persistent organic pollutants (POPs) and metals, i.e., bisphenol A (BPA), phthalates and polybrominated-diphenyl-ethers (PBDEs), but did not take into consideration their compounding effects. Finally, Hinault et al., 2023 [21] affirmed that it remains difficult to form a comprehensive view on the causal relationship between EDCs and diabetes (both T1DM and T2DM) and that further experiments are required.

## 5. Conclusions

Based on the data examined, it can be concluded that stronger association was found between persistent organic pollutants (POPs), phthalates and bisphenols, mostly BPA, and the risk for development of T1DM, mainly through reduced insulin sensitivity, autoimmunity of pancreatic β-cells, oxidative and inflammatory stress and epigenetic modulation. It is obvious that further research is required to bear out the important role of ECDs in the pathophysiology of DM in general and, specifically, T1DM; studies need to consider the compound and accumulating effects of many disruptors at a time. An important setback to this analysis was the limited data found compared to the wide variety of ECDs described; however, based on the data collected and the quality assessments performed, the studies found are credible and valid enough to support the correlation and linkage between endocrine disruptors and T1DM.

## Figures and Tables

**Figure 3 ijms-25-10111-f003:**
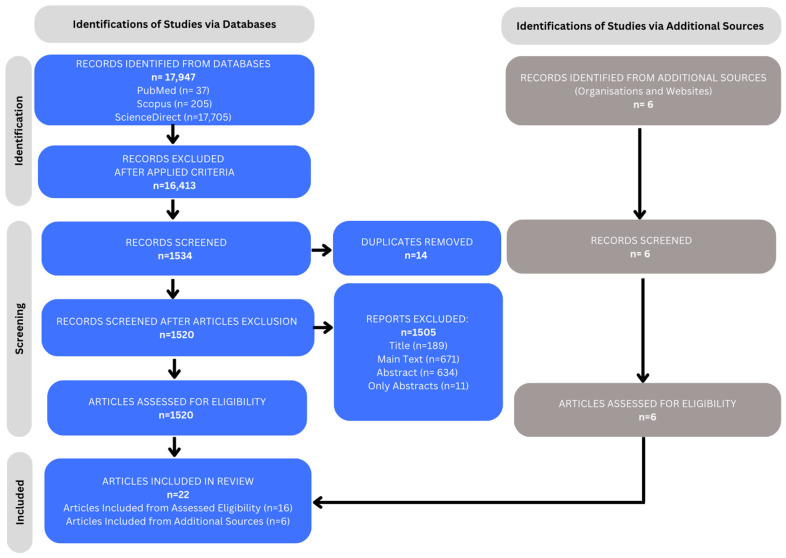
PRISMA flow diagram.

**Table 1 ijms-25-10111-t001:** Population, Intervention, Comparator, Outcome.

PICO	
P	Children and Adolescents and Young Adults (10–22 years old)
I	Endocrine-Disrupting Chemicals
C	Diabetes Mellitus Type 1 (T1DM)
O	Endocrine Disrupting Chemicals as Risk Factors of Diabetes Mellitus Type 1 (T1DM)

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
