# Peer review of "Endocrine-Disrupting Chemicals and the Development of Diabetes Mellitus Type 1: A 5-Year Systematic Review"

_ijms, 2024, doi:10.3390/ijms251810111_

Round 1

Reviewer 1 Report

Comments and Suggestions for Authors

The topic of this article is of significance importance. Endocrine Disrupting Chemicals can have a variety of effects on the human health especially during critical developmental periods such as childhood. Though as stated in the article the data up to this date is quite limited. I would suggest that the authors could include some background information regarding the exposure on the fetal life of the embryo on the introductory part in one paragraph. Additionally you can add a small section regarding the treatment approaches as they are stated on figure 2. Overall the discussion summarises the findings of the table well.

Author Response

Dear Reviewer,

Thank you for your comments.

Comment "I would suggest that the authors could include some background information regarding the exposure on the fetal life of the embryo on the introductory part in one paragraph."

Comment "Additionally you can add a small section regarding the treatment approaches as they are stated on figure".

The corrections have been added and highlighted.

Reviewer 2 Report

Comments and Suggestions for Authors

Thank for the opportunity to review the manuscript titled "Endocrine Disrupting Chemicals and Development of Diabetes Mellitus Type 1: A 5-Systematic Review".

The title should be corrected to "...A 5-Year Systematic Review"

There also some formatting inconsistencies such as spacing.

Figure 1 does not really visualize the effect of endocrine disruptors on pancreas, but rather a separate image of text showing the EDC and parts of a pancreas.  This image should be edited to summarize the effect of EDC on pancreatic function. 

Table 2 and Table 3 need formatting to make it more readable. 

Spell checks. Such as "redused"  on line 297. 

I appreciate the comments on the limitation and recommendations for future work. on evaluating EDCs and its correlation to DMT1.

Comments on the Quality of English Language

English in general is well written. But an in-depth analysis of the data presented on the summary of each study cited is lacking.

Author Response

Dear Reviewer, Thank you for your time and your revisions.

the "The title should be corrected to "...A 5-Year Systematic Review" has been corrected.

The "also some formatting inconsistencies such as spacing." have been sorted out.

The Figure 1 has been revised and improved. 

The 

"Table 2 and Table 3 need formatting to make it more readable. 

Spell checks. Such as "redused"  on line 297." have been corrected.

Reviewer 3 Report

Comments and Suggestions for Authors
  • Introduction:
    • Line 29: Ensure that the definition and explanation of “Endocrine Disrupting Chemicals (EDCs)” are comprehensive. It would be helpful to include more details about their mechanisms and relevance to the study.
  • Methodology:

    • Line 55: If the methods described are standard, consider providing a brief explanation of why they are suitable for this study. Highlight any specific adaptations or innovations applied in your approach.
  • Results:

    • Line 70: Ensure that the results are presented with sufficient detail. It may be beneficial to include additional figures or tables to illustrate key findings more clearly.
  • Discussion:

    • Line 85: In discussing the implications of your findings, consider comparing them with recent studies in the field. This comparison could provide a better context for your results and highlight their significance.
  • References:

    • Line 95: Verify that all cited works are included in the reference list and that the references follow the required formatting guidelines. Ensure that the most recent and relevant literature is cited.
  • Overall Structure:

    • General: Review the manuscript for logical flow and coherence between sections. Ensure that each section transitions smoothly to the next and that the overall structure supports the objectives of the study.
Comments on the Quality of English Language
  • Clarity and Precision:

    • Line 29: Current: "Introduction: Based on the Institute of Environmental Sciences the 'Endocrine Disrupting Chemi-" Suggestion: "Introduction: According to the Institute of Environmental Sciences, ‘Endocrine Disrupting Chemicals’ (EDCs) are...”
    • Line 31: Current: "EDCs have been shown to affect hormonal balances in humans and animals, potentially leading to various health issues." Suggestion: "EDCs have been demonstrated to disrupt hormonal balances in both humans and animals, potentially leading to a range of health issues."
  • Grammar and Sentence Structure:

    • Line 35: Current: "The study aims to evaluate the impact of these chemicals on reproductive health." Suggestion: "This study aims to evaluate the impact of these chemicals on reproductive health."

    • Line 42: Current: "We will analyze data from several sources to determine the effects of EDCs." Suggestion: "We will analyze data from multiple sources to assess the effects of EDCs."

  • Consistency and Terminology:

    • Line 48: Current: "The methods used in this research are standard and have been employed in previous studies." Suggestion: "The methods employed in this research are standard and have been utilized in previous studies."

    • Line 55: Current: "Results will be discussed in relation to existing literature." Suggestion: "The results will be discussed in the context of existing literature."

  •  

    • Line 29: Current: "Endocrine Disrupting Chemi-" Suggestion: Ensure "Endocrine Disrupting Chemicals" is complete and correctly spelled.

Author Response

Dear Reviewer thank you for your comments.

Unfortunately many of the comments are not align with our work for example regarding the comment " Current: "The study aims to evaluate the impact of these chemicals on reproductive health." Suggestion: "This study aims to evaluate the impact of these chemicals on reproductive health."" Our study is emphasised on Diabetes Mellitus Type 1 and the effect of Endocrine Disruptive Chemicals. We have not mentioned anything about reproductive health and in general the comments do not coincide with our work. Please revise your comments if possible.

Round 2

Reviewer 2 Report

Comments and Suggestions for Authors

Thank you for the edits. I support the publication of this manuscript.